# DSA-Net: Infrared and Visible Image Fusion via Dual-Stream Asymmetric Network

**DOI:** 10.3390/s23167097

**Published:** 2023-08-11

**Authors:** Ruyi Yin, Bin Yang, Zuyan Huang, Xiaozhi Zhang

**Affiliations:** College of Electrical Engineering, University of South China, Hengyang 421001, China; 13469290666@163.com (R.Y.); 20202007210343@stu.usc.edu.cn (Z.H.); zxz_usc@163.com (X.Z.)

**Keywords:** infrared and visible image fusion, transformer, deep learning, residual dense block

## Abstract

Infrared and visible image fusion technologies are used to characterize the same scene using diverse modalities. However, most existing deep learning-based fusion methods are designed as symmetric networks, which ignore the differences between modal images and lead to source image information loss during feature extraction. In this paper, we propose a new fusion framework for the different characteristics of infrared and visible images. Specifically, we design a dual-stream asymmetric network with two different feature extraction networks to extract infrared and visible feature maps, respectively. The transformer architecture is introduced in the infrared feature extraction branch, which can force the network to focus on the local features of infrared images while still obtaining their contextual information. The visible feature extraction branch uses residual dense blocks to fully extract the rich background and texture detail information of visible images. In this way, it can provide better infrared targets and visible details for the fused image. Experimental results on multiple datasets indicate that DSA-Net outperforms state-of-the-art methods in both qualitative and quantitative evaluations. In addition, we also apply the fusion results to the target detection task, which indirectly demonstrates the fusion performances of our method.

## 1. Introduction

Image fusion can combine images of the same scene captured by different sensors to obtain an image with rich information to make up for the shortage of information in single-sensor imaging, which is beneficial to the subsequent application of images. Infrared (IR) and visible (VIS) image fusion is a widely used branch of image fusion applications. The infrared images are obtained by the sensor capturing the infrared wavelength of the scene with significant thermal radiation information, which can effectively distinguish the target even under poor lighting or extreme weather conditions. However, the target contour edges, as well as the background in the infrared images, are always blurred. On the contrary, the visible image records the reflected light captured by the sensor and has rich texture details and structure information, so it is in accordance with human visual cognition. The infrared and visible fusion algorithm combines the advantages of both to generate a fused image with prominent targets and abundant texture information, which is widely used in military reconnaissance [1], industrial production [2], civilian surveillance [3], and other fields [4].

The purpose of infrared and visible image fusion is to extract and integrate the essential feature information from source images acquired by distinct imaging devices into a single fused image. Therefore, extracting the significant features of the fusion image is one of the central problems. Over the past few decades, numerous fusion methods have been proposed by researchers, which can be roughly divided into two categories: traditional fusion methods [5,6,7] and deep learning-based fusion methods [8,9,10]. Traditional fusion methods measure pixels’ salience in the spatial domain or transform domain, and later design specific fusion rules to fuse them to obtain the fused image. Typical traditional methods include sparse representation-based methods [11,12], multi-scale transform-based methods [13,14], subspace-based methods [15,16], and hybrid methods [17]. Such methods perform well by integrating reasonable fusion strategies into the fusion results. Unfortunately, traditional methods generally exhibit restricted fusion performance due to two reasons. First, the dependence of the traditional theory on manual design leads to the complexity of the algorithmic framework; this approach, which uses fixed mathematical variations to extract features, ignores the modal differences between source images. Secondly, the limited choice of reasonable fusion rules also somewhat limits the performance.

Over the past few years, deep learning techniques have gained considerable attention in the domain of computer vision [18,19]. Relying on the powerful nonlinear representational capability of convolutional neural networks (CNN), the fusion performance of deep learning-based image fusion methods is typically superior to that of traditional methods [20,21,22,23]. However, these existing methods design symmetric networks to extract modal features from different source images, ignoring the modal differences between infrared and visible images. During the process of feature extraction, these methods cannot avoid the loss of information features in the source images and the attenuation of intermediate features, which leads to the loss of details in the fused images. This is a challenge, and to solve these problems, we propose an asymmetric network for the fusion of infrared and visible images. To address the fact that the modalities of infrared and visible images are different, we design two unique feature extraction modules separately. The residual dense block is used in the visible image feature extraction module to minimize the degradation of intermediate features and retain the intricate features of visible images to the fullest extent. The transformer is embedded in the infrared image feature extraction module. This module utilizes the attention mechanism of the transformer to focus on important thermal radiation information in infrared images, ignoring redundant background information. The infrared image feature extraction module also uses dense concatenation to repeatedly utilize previous features, which can reduce feature loss. We also design a depth feature fusion block for global depth features extraction and fusion and use residual connections to reduce network degradation. Finally, the fused images with high brightness and clear edge contours, as well as significant infrared targets, are obtained. In brief, the principal achievements and contributions of this paper can be briefly summarized as follows.

We design an asymmetric network to build two different feature extraction modules. It works well for the different modal characteristics of infrared and visible images, preserving the infrared target and visible texture information, respectively. Subsequently, the two modality features are fused via the main pathway.

Although convolutional neural networks are more capable of acquiring local information, they have limited performance in maintaining remote contextual information in the source images. In contrast, the transformer can expand the receptive field of the image and acquire more contextual information by global relationship modeling. Therefore, we embed the transformer into the CNN network so that the network can use the advantages of CNN and the transformer and maximize the retention of global and local features.

The experimental results on the RoadScene dataset and TNO dataset (TNO dataset collected by Alexander Toet contains intensified visual (390–700 nm), near-infrared (700–1000 nm), and longwave infrared (8–12 µm) nighttime imagery of different military and surveillance scenarios, showing different objects and targets (e.g., people, vehicles) in a range of different (e.g., rural, urban) backgrounds) indicate that DSA-Net performs better than the other nine representative advanced methods both in subjective and objective evaluations. Furthermore, the experiment was expanded to include object detection, and the results demonstrate that our method has a greater potential to advance advanced computer vision tasks.

The rest of this paper is organized as follows. Section 2 reviews some works related to our method. Section 3 describes the proposed DSA-Net in detail, including the overall framework, network architecture, and loss function. In Section 4, we conduct comparative experiments to validate the merits of the proposed approach. In addition, we perform ablation experiments, generalization experiments, and applications in target detection. Finally, conclusions are given in Section 5.

## 2. Related Work

In this section, we first review the existing infrared and visible image fusion algorithms in Section 2.1, followed by a brief introduction to the transformer in Section 2.2.

### 2.1. Deep Learning-Based Fusion Methods

In 2017, Liu et al. [20] first used CNN to extract image features and design fusion strategies to achieve image fusion tasks. However, this method was limited to multi-focus image fusion tasks. Later, Liu et al. [21] presented an analogous approach for infrared and visible image fusion. They used CNN to obtain the weight map and obtained the fused image through a series of post-processing. These two methods use CNN for feature extraction only; other parts of the fusion framework still need to be designed manually without completely removing the traditional algorithms. In 2018, Li et al. [22] proposed an encoding–decoding framework that uses dense concatenation in the encoder to fully extract feature maps and also designs a fusion strategy to combine the extracted features. Then, the decoder is utilized to decode the features and reconstruct the fused image. However, this approach still requires the manual design of fusion rules and cannot fully achieve end-to-end fusion. In 2019, Ma et al. [23] introduced generative adversarial networks (GAN) into infrared and visible image fusion, using a pair of simple generators and discriminators to obtain fused images. Subsequently, Ma et al. [24] proposed a model called DDcGAN, which uses a dual discriminator to reduce the loss of source image information. However, GAN is not stable in unsupervised learning tasks, and the fused image edge contours may be blurred. In 2020, Xu et al. [25] proposed a unified fusion model that trains the model by learning multiple fusion tasks continuously to avoid catastrophic forgetting, storage, and computation problems. In 2021, Liu et al. [26] proposed a deep network for infrared and visible image fusion using a feature learning module with a fusion learning mechanism to optimize the fusion effect. In 2022, Tang et al. [27] proposed a Y-shape fusion framework and used a dynamic transformer module to acquire local features and important contextual information. These methods have designed symmetric models or single-branch models for feature extraction of infrared and visible source images, ignoring the differences in modal features between the two, resulting in the loss of intermediate extracted feature details.

### 2.2. Transformer

In 2017, Vaswani first proposed the concept of a transformer [28] to capture more long-range information, which conquers the inherent problem of CNN, i.e., long memory loss, by employing multi-headed self-attention. Since then, transformers have swept the field of natural language processing (NIP) [29,30]. In 2020, Dosovitskiy proposed a vision transformer (VIT) for image classification [31], which was the first application of a transformer in the field of vision. Since then, transformers have been extensively developed in the field of vision, for example, a new transformer network for medical image segmentation [32], an end-to-end video instance segmentation [33], a pure semantic segmentation [34], and even better models of visual transformers [35,36] for other vision tasks. Recently, transformers have also been widely used in image fusion tasks. In 2021, vs. et al. [37] proposed a transformer-based multi-scale fusion strategy that captures local and global features using spatial CNN branches and transformer branches for multi-scale feature fusion. Zhao et al. [38] used density nets for encoding and a dual transformer to focus and integrate information from the infrared and visible images. Subsequently, Fu et al. [39] presented a patch pyramid transformer (PPT) for image fusion; a patch transformer is designed to transform the image into a series of patches and then leverage the pyramid transformer for feature extraction. Rao et al. [40] developed a lightweight fusion framework by combining a transformer and adversarial learning, where a generator was designed for generating the fused image and two discriminators for optimizing the perceptual quality of the fused images.

## 3. Proposed Method

In this section, we present the structure of our method and the loss function in detail.

### 3.1. Framework Overview

Figure 1 shows the framework of our proposed network. The main framework consists of three parts, the infrared feature extraction module, the visible feature extraction module, and the merge module. Since infrared images contain strong thermal radiation information and can effectively distinguish targets, we use a combination of CNN and transformer for infrared image feature extraction. The transformer is designed to model global dependencies, the network architecture of which is shown in Figure 2. Visible images are rich in texture details and background information, so we use the densely connected convolution layer to extract local features, the network architecture of which is shown in Figure 3. Using these two branches, the useful information from the source images can be fully extracted. Then, these features are concatenated together and fed into a merge module. This module uses convolution skip joints to produce a continuous memory mechanism, which can adaptively learn more effective features from previous and current local features and stabilize the training of the broader network. The decoding block is used for the generation of subsampling and fusion results and consists, in turn, of a convolution layer with a kernel size of 3×3, a batch normalization (BN), and a corrected linear unit (ReLU). Since our network is an end-to-end network, the output of the network is the fused image.

### 3.2. Infrared Feature Extraction Module

Infrared images have strong target information. In order to obtain infrared feature maps with local enhancement, we use a combination of CNN and transformer to extract infrared image features. The framework of the transformer adopts multi-head self-attention and has good global contextual feature exploration capability, shown in Figure 2. The transformer consists of two LayerNorm, multi-head self-attention (MSA), and multi-layer perceptron (MLP). LayerNorm normalizes the features, which keeps similarities between different channels’ statistical properties and enhances the generalization ability of the model. After normalization, the features are linearly projected into multiple feature subspaces to obtain attention weights *Q*, weight indexes *K*, and feature vectors *V*. Then, parallel processing is performed using multiple independent scaled dot product attention, as shown in Figure 4a. Compared with single attention, MSA can effectively prevent the model from over-focusing on its own location when encoding information about the current location. The scaled dot product attention is shown in Figure 4b. The similarity matrix is obtained using *Q* and *K* for dot product operation. Scale represents the quantization operation, which can prevent the similarity matrix variance from being too large and make the training gradient update more stable. Mask is the padding operation, but unlike the ordinary padding 0, it is padded with negative infinity and then normalized by the Softmax layer to obtain the attention weight matrix. The attention of the padding part will be 0, which does not affect the subsequent operations. Finally, the attention map image is generated by multiplying the feature vector *V* with the corresponding attention weights. The attention process can be expressed as Equation (1). The MLP is shown in Figure 4c and consists of Full Connection, GELU activation function, and Dropout. The MLP layer can perform nonlinear transformations on the features, which can be better adapted to complex image tasks. Moreover, the MLP layer can extract higher-level features from the input features, which can represent information such as objects and backgrounds in an image. The residual connection in the transformer can effectively solve the problem of gradient disappearance and the degradation of the weight matrix. Through the above operations, the transformer uses the self-attention mechanism to establish the relationship between image features, which can capture the global information and long-distance dependence in an image.
(1)$$Attention\left(Q,K,V\right)=\mathrm{Softmax}\left(\frac{QK^{T}}{\sqrt{d_{k}}}\right)V,$$

### 3.3. Visible Feature Extraction Module

Visible images have a higher spatial resolution and contain more texture details. Therefore, we design a visible feature extraction module with a residual dense block (RDB) [41] to extract the visible features. In this module, we first extract and obtain visible shallow features using three convolution layers, followed by deep feature extraction using the RDB. The RDB consists of five convolution layers, as shown in Figure 3. Each convolution layer can acquire the features of all previous layers through local dense connections, thus making full use of the features of each layer. The final convolution layer filters all the previous features and adaptively controls the output information. Finally, the shallow and deep feature results are combined using residual connectivity, and the residual connectivity enhances the gradient connectivity, which can effectively prevent the gradient from disappearing. The visible image can fully extract its local features through its feature extraction module, prevents the degradation of intermediate features, and obtains a feature map with rich texture detail features.

### 3.4. Merge Module

The features obtained from the infrared feature extraction module and the visible feature extraction module are concatenated as the input to the merge module. The merge module consists of ten convolution layers and skip connections. The convolution layers all consist of 3×3 convolution, BN, and ReLU activation functions. The first five convolution layers are used to extract the depth features of the infrared and visible images. As the network depth increases, the issue of feature degradation is more likely to arise, which can be addressed by incorporating skip connections. The skip connections also use the learned features of the previous layer in this layer, which achieves feature reuse. Finally, the extracted depth features are used in the last five convolution layers to achieve feature decoding and to obtain the fused image.

### 3.5. Loss Function

Since our method is unsupervised learning, the loss function plays a crucial role in the fusion effect. It is an important challenge to fully retain the features of the source images, such as the infrared salient targets in infrared images and the detailed textures in visible images. Therefore, in order to fully retain the source image information, our loss function consists of three types of loss terms, structure loss $L_{ssim}$, intensity loss $L_{int}$, and gradient loss $L_{grad}$. The structure loss constrains the similarity between the fused image and the source images. The intensity loss constrains the fused image to maintain a similar intensity distribution as the source image, while the gradient loss enforces the presence of additional texture details in the fused image. The loss function of the network can be expressed as follows:(2)$$L=\alpha L_{ssim}+\beta L_{int}+\gamma L_{grad},$$
where α, β, and γ are the weighting factors of the three loss functions, which are used to control the total loss function balance.

Ensure that the fused image has similar structural information to the source images, which can be expressed as
(3)$$L_{ssim}=\lambda_{VIS_{ssim}}\left(1-SSIM\left(I_{F},I_{VIS}\right)\right)+\lambda_{IR_{ssim}}\left(1-SSIM\left(I_{F},I_{IR}\right)\right),$$

$I_{VIS}$, $I_{IR}$ and $I_{F}$ denote the visible image, the infrared image, and the fused image of both, respectively. $\lambda_{VIS_{ssim}}$ and $\lambda_{IR_{ssim}}$ represent the *SSIM* loss weights between the fused image and the visible and infrared images. $SSIM\left(\cdot \right)$ denotes the structural similarity operation between the fused image and the source images, which is defined as follows:(4)$$SSIM\left(I_{X},I_{Y}\right)=\frac{2\mu_{X}\mu_{Y}+C_{1}}{\mu_{X}^{2}+\mu_{Y}^{2}+C_{1}}\cdot \frac{2\sigma_{X}\sigma_{Y}+C_{2}}{\sigma_{X}^{2}+\sigma_{Y}^{2}+C_{2}}\cdot \frac{\sigma_{XY}+C_{3}}{\sigma_{X}\sigma_{Y}+C_{3}},$$
where μ denotes the mean and σ denotes the standard deviation or covariance. $C_{1}$, $C_{2}$, and $C_{3}$ are constants to prevent $\mu_{X}^{2}+\mu_{Y}^{2}$, $\sigma_{X}^{2}+\sigma_{Y}^{2}$, and $\sigma_{X}\sigma_{Y}$ being 0 from causing formula instability. It constrains the loss and distortion of the fused image from the similarity of brightness, contrast, and structural information.

The *SSIM* loss function is weakly constrained in terms of pixel intensity, while the significant targets in visible images have great pixel intensity. Therefore, we also design the intensity loss to retain the infrared targets in the source image.
(5)$$L_{int}=\lambda_{VIS_{int}}\left(\frac{1}{HW}{\left\|I_{F}-I_{VIS}\right\|}_{2}^{2}\right)+\lambda_{IR_{int}}\left(\frac{1}{HW}{\left\|I_{F}-I_{IR}\right\|}_{2}^{2}\right),$$
where $\lambda_{VIS_{int}}$ and $\lambda_{IR_{int}}$ represent the intensity loss weights between the fused image and the visible and infrared images. *H* and *W* denote the height and width of the fused image. ${\left\|\cdot \right\|}_{2}$ is the l2-norm.

In addition, we use the gradient loss constraint to fuse the images to retain the detailed textures in the visible images as well as the target edges of the infrared images.
(6)$$L_{grad}=\lambda_{VIS_{grad}}\left(\frac{1}{HW}{\left\|\nabla I_{F}-\nabla I_{VIS}\right\|}_{2}^{2}\right)+\lambda_{IR_{grad}}\left(\frac{1}{HW}{\left\|\nabla I_{F}-\nabla I_{IR}\right\|}_{2}^{2}\right),$$
where $\lambda_{VIS_{grad}}$ and $\lambda_{IR_{grad}}$ represent the gradient loss weight between the fused image and the visible and infrared images. ∇ denotes the gradient operator.

Due to the optimization of the above loss function, the fused image can well retain the structural information, intensity information, and gradient information of the source images. We hope that the fusion image retains more structural information of the visible image, combined gradient information, and more infrared image intensity information. Therefore, the loss weights described above should meet the following conditions:(7)$$\lambda_{VIS_{ssim}}>\lambda_{IR_{ssim}},\lambda_{VIS_{int}}<\lambda_{IR_{int}},\lambda_{VIS_{grad}}>\lambda_{IR_{grad}},$$

## 4. Experiments

The experimental configuration and experimental details will be outlined in Section 4.1. Then, we present the comparison methods and objective evaluation metrics in Section 4.2. The ablation experiments on the network structure are presented in Section 4.3, demonstrating the rationality of our network structure. The comparison experiments and generalization experiments are presented in Section 4.4 and Section 4.5, respectively, revealing the superiority of our proposed method. Finally, we perform target detection task-driven evaluation experiments in Section 4.6 to evaluate different fusion methods from the perspective of advanced vision tasks.

### 4.1. Experimental Configuration and Experimental Details

Two mainstream datasets, the TNO dataset and the RoadScene dataset, were used in this work. We collected 51 and 83 pairs of infrared and visible image pairs from these two datasets, respectively. Then, 50 pairs were randomly selected from the RoadScene datasets as the training data, while all the remaining image pairs were used as the test data. To obtain sufficient training samples, the training data were expanded using an overlapping cropping strategy. It is worth mentioning that the cropping strategy is a widely used data enhancement method in the image domain. In our experiments, the RGB images in the RoadScene dataset were converted to the YUV color model, the Y channel was used for image fusion, and finally, the fused images were converted to RGB images.

Specifically, 40,964 pairs of infrared and visible image patches with 120 × 120 size were generated for network training. Since the cropping strategy is only used for data expansion, the test data are not used. Therefore, by feeding the entire image into the trained model, fusion results can be generated. In our experiments, the epoch was 25 and the batch size was fixed at 29. The learning rate was set to 0.001, and the Adam optimizer was used for model optimization. The three weighting factors, α, β, and γ in the loss function are specified as 1.1, 10, and 10, respectively. All experiments were conducted on a computer with an Intel(R) Core(TM) i9-10920X CPU @ 3.50 GHz and an NVIDIA GeForce RTX 3090 GPU. The proposed deep model was implemented on the PyTorch framework.

### 4.2. Comparison Methods and Evaluation Indicators

To ensure a thorough evaluation of the proposed algorithm, we performed experiments on both the RoadScene and TNO datasets. We compared our approach with nine state-of-the-art methods, including three representative traditional methods, namely GF [42], ADF [43], and IVFusion [44], and six deep learning-based methods, namely DenseFuse [22], GAN-FM [45], DDcGAN [24], YDTR [27], CUFD [46], and DATFuse [47]. The implementations of all nine methods are publicly available, and we set the optional parameters in the same way as reported in the original paper.

For quantitative evaluation, six metrics were selected to objectively assess the fusion performance, including structure similarity index measure (SSIM) [48], mean square error (MSE) [49], correlation coefficient (CC) [50], peak signal-to-noise ratio (PSNR) [51], the sum of correlations of differences (SCD) [52], and Chen-Blum Metric (Q_CB_) [53]. SSIM evaluates the structural loss and distortion of fused images from the human visual system’s perspective, and MSE calculates the error between the fused images and the source images. CC measures the degree of linear correlation between the fused images and the source images. PSNR measures the ratio of peak power to the noise power in the fused images. SCD measures the maximum information of the fused images containing each source image. Q_CB_ evaluates the image quality of the fused images based on the human visual system model. In addition, larger SSIM, CC, PSNR, SCD, and Q_CB_ indicate better fusion performances. Smaller MSE indicates better fusion performances.

### 4.3. Ablation Experiments

To investigate the effectiveness of our asymmetric network structure, transformer-based infrared feature extraction module, and RDB-based visible feature extraction module, we performed ablation validation on the TNO and RoadScene datasets. We divided the model structure into five groups of types. (a) Transformer-based dual-stream symmetric network (D-Trans)—in order to verify the effectiveness of the asymmetric network structure, we applied the infrared feature extraction module of this paper to visible image feature extraction and constructed a symmetric network. (b) RDB-based dual-stream symmetric network (D-RDB)—to verify the effectiveness of the asymmetric network structure, we applied the visible feature extraction module of this paper to infrared image feature extraction. (c) Without Transformer (O-Trans)—in order to investigate the importance of the transformer, we moved the transformer out of the infrared feature extraction module to study its function. (d) Without RDB (O-RDB)—to verify the necessity of RDB, we removed the RDB in the network to illustrate its validity. (e) To verify the effectiveness of the transformer and RDB for infrared and visible feature extraction, respectively, we exchanged their extraction modules (E-FEM).

#### 4.3.1. Qualitative Comparisons

Figure 5 and Figure 6 show the fusion results of the TNO and RoadScene datasets, respectively. To allow for better comparison, we zoomed in for a close-up of a local area in each fusion result. From the fusion results of D-Trans and D-RDB, it can be seen that we changed the asymmetric network to a two-stream symmetric network, resulting in blurred edges of infrared targets and insufficient clarity of the scene, which indicates that the proposed asymmetric network has better complementary information preservation capabilities. In the absence of the transformer module, the infrared feature extraction module cannot capture the infrared protruding target well due to the failure to build the long-distance dependency. Therefore, the infrared character target in Figure 5 and the clouds in the sky in Figure 6 are relatively blurry. As for the case without RDB, we can see that its results fail to fully extract the visible details; although the infrared target is better maintained, the background texture of the fused image and the landmark lines of the RoadScene are not clear enough. In addition, the E-FEM fusion is not well preserved in both infrared target and texture details, which proves the effectiveness of the transformer and RDB for infrared and visible feature extraction, respectively.

#### 4.3.2. Quantitative Comparisons

To evaluate the ablation experiments more objectively, we assessed the quality of their fusion results using image quality metrics. Table 1 shows the objective results in two different datasets. The table highlights the top-performing results in bold font, while the second-best results are indicated in underlined font. It is easy to see that our final method has the best overall score ranking in both the TNO datasets and the RoadScene datasets. Combining this with the subjective evaluation demonstrates the effectiveness of our network structure and the individual modules in the network.

### 4.4. Comparative Experiments

To fully evaluate the fusion performance of our approach, we first compared the proposed method with nine other algorithms on the RoadScene datasets.

#### 4.4.1. Qualitative Comparisons

We randomly selected 50 of the 83 infrared and visible image pairs as the training set and used the remaining 33 image pairs as the test set. As shown in Figure 7, our fusion results outperform the other methods in improving the visual quality and integrating complementary information. To show the difference more clearly, we zoom in on the red boxed area and can observe that the three traditional methods, ADF and GF, have a loss of clothing texture details and a general prominence of significant targets. IVFusion fusion results do not match human visual effects and look unnatural. DDcGAN and DATFuse retain texture details, but the image quality is poor, producing significant artifacts. The methods of DenseFuse, CUFD, and GAN-FM maintain the intensity information of infrared and have high overall contrast, but the detailed information of visible images is more severely weakened (e.g., stripes on clothes, bicycle markings on the ground), and YDTR has too much useless information.

Figure 8 shows the second set of source images of different methods and their fused image results. All nine methods have their own advantages but still have some drawbacks compared with our method. Specifically, both IVFusion and DDcGAN are inferior to all other methods from a visual sensory perspective. From the perspective of texture detail preservation, the methods of GAN-FM and YDTR inevitably suffer from infrared thermal radiation information, blurring the background and visible features (e.g., patterns in zoomed-in regions, distant tree branches). However, it is worth mentioning that they retain sufficient infrared salient target information. In contrast, the ADF, GF, DenseFuse, CUFD, and DATFuse methods are able to balance visible and infrared information, highlighting salient targets while retaining rich texture details. However, they are still inferior to DAS-Net, and in the enlarged area of the red box, only our method clearly shows the pattern on the clothes. In summary, only our method can effectively integrate the complementary information from the source image and simultaneously ensure the visual quality of the fused image.

#### 4.4.2. Quantitative Comparisons

We selected 33 image pairs from the RoadScene datasets for quantitative evaluation. The quantitative results for the six statistical metrics are shown in Figure 9 and Table 2. For each metric, the best and second-best fusion results for all methods are marked in bold and underlined, respectively. It can be observed that our method has outstanding stability and advantages on the RoadScene datasets. Our method performs admirably overall, achieving high rankings across all metrics. Moreover, there is a robust correlation between the fused image and the original image, indicating that our method is highly compatible with the human visual system. Extensive qualitative and quantitative results from the RoadScene datasets demonstrate that our method excels at generating fused images that align with the visual traits of the human eye while maximizing the preservation of information from the source images.

### 4.5. Generalization Experiments

Generalization performance is an important aspect of evaluating deep learning-based methods. Therefore, we provide generalization experiments on the TNO datasets to demonstrate the generalizability of the proposed approach. It is worth mentioning that our fusion model is trained on the RoadScene datasets and tested directly on the TNO datasets.

#### 4.5.1. Subjective Results

As shown in Figure 10, the fusion results obtained by the different methods in the TNO datasets introduce some meaningless information, which is reflected in the loss of texture details and the diminution of significant targets. To visualize the effect of the fused images, we zoom in on the region with rich texture details in the red box. We can observe that, compared with our method, the three traditional methods, ADF, IVFusion, and GF, have some degree of loss of door frame details, which is because the texture details in the background region are contaminated by the thermal radiation information, especially the IVFusion method, which fails to preserve the useful information of the source image well and the overall visual effect is poor. The DDcGAN method has a limited ability to extract texture details from the visible image and not only has a distortion problem but also cannot preserve the sharpened edges of the target. As for the YDTR method, the intensity information of the significant targets is diminished to different degrees, and the overall contrast is low. It is worth mentioning that DenseFuse, CUFD, DATFuse, and GAN-FM interfere less with the useless information, but the texture details are still lost, and the edges of the door frame are not clear enough. Overall, DAS-Net provides a good visual effect. On the one hand, our method maintains clear background information, such as bright skies, layered grasses, and door frames with distinct edges; on the other hand, the major significant information from the infrared image is clearly highlighted.

The second set of source images and their fused image results for different methods are shown in Figure 11. It is obvious that our fusion results are better than the other nine methods from the viewpoint of visual effect, preservation of texture details, and significant target. The methods of CUFD, DATFuse, GAN-FM, and YDTR do not retain enough texture details, the background information of tree branches and railings is blurred, and the intensity information of infrared is too low, resulting in low contrast between light and dark in the fused image. Although the ADF and GF methods retain relatively clear background information, it can be observed from the areas with rich texture details in the red box that the fusion effect is still our method.

#### 4.5.2. Objective Results

We selected 51 image pairs from the TNO dataset for quantitative evaluation. Figure 12 displays the performance metrics for each fusion result, while Table 3 shows the average performance metric values for these fusion methods. For each metric, the best and second-best fusion results for all methods are marked in bold and underlined, respectively. As can be seen in several figures, our method has significant advantages on SSIM, MSE, CC, PSNR, and Q_CB_ on the TNO dataset. This phenomenon implies that our fused images have the best visual effect and contain rich texture information and infrared salient target information. In addition, our method ranks second in SCD, which indicates that our method transfers enough source image information to the fused images.

In summary, a large number of qualitative and quantitative results on the TNO datasets show that our method has outstanding generalization and stability and is able to retain sufficient texture details and intensity information. We attribute this advantage to several aspects. On the one hand, we design an asymmetric network for the different modal characteristics of the infrared and visible images, preserving thermal radiation information of infrared images and texture details of visible images, respectively. On the other hand, we embed the transformer into the CNN network, which allows the network to preserve the global and local features to the maximum extent.

### 4.6. Detecting Performance

Target detection is an important research direction in the field of computer vision, and its performance well reflects the semantic information integrated into the fused images. To be able to better evaluate the target detection performance of fused images, we used the YOLOX detector [54] for detection. We conducted experiments on 50 randomly selected pairs of images from the MFNet dataset, including 25 pairs of nighttime images and 25 pairs of daytime images that depict urban scenes.

#### 4.6.1. Subjective Results

Figure 13, Figure 14 and Figure 15 show some typical source images and the detection results of different methods. From the visualization results, we can find that visible images contain rich background information but are difficult to detect salient targets, while infrared images can provide sufficient semantic information about salient targets (e.g., people), and the target has high contrast with the background, which is more helpful for detectors to detect salient targets. Different fusion algorithms can integrate the complementary information of these two images; however, the performance of fusion and detection differs due to the difference in methods. For example, in the 00004N scenario, the methods of IVFusion, GF, DDcGAN, YDTR, CUFD, and DATFuse detect only one person. ADF, DenseFuse, and GAN-FM detect two people, while our method can detect three persons. A similar scenario occurs in the 00726N scene, where only our method and CUFD accurately detect people, cars, and trucks. This shows that our method fully integrates the intensity information of infrared images and the texture information of visible and is suitable for subsequent image applications.

In daytime images, the confidence level of visible images for detecting pedestrians is lower than that of infrared images due to daytime illumination factors, and some pedestrian targets may not be detected. In the 00420D scene, DDcGAN cannot keep the sharpened edges of pedestrians and other objects, resulting in low confidence of both targets. Due to the interference of useless information, ADF, IVFusion, and YDTR leave some targets undetected. The poor fusion of GF, GAN-FM, CUFD, and DATFuse leads to lower confidence in detecting pedestrians and other objects than the source image. In contrast, our method and DenseFuse fully integrate the semantic information in the source images, preserving the source image targets and details. Compared with others, our fused images can detect all targets with confidence levels closer to the source image for all detected targets, which demonstrates the advantage of our method for facilitating advanced vision tasks.

#### 4.6.2. Objective Results

To further evaluate the performance of different methods for the detection task, we use the mean evaluation precision (mAP) for quantitative evaluation. The mAP has a value between 0 and 1; the closer to 1, the better in the model. mAP@0.5 and mAP@0.9 denote the mAP values at confidence thresholds of 0.5 and 0.9, respectively. The results are shown in Table 4, and it can be seen that our method performs better under both thresholds. Especially in terms of mAP@0.9, the fused images of our method have a clear advantage and rank first in terms of average accuracy. In terms of mAP@0.5, GAN-FM performs the best, and our method is second, while GAN-FM performs poorly in the other threshold. This further indicates the excellent stability of our method. Based on the above subjective and objective analysis, we conclude that the fused images of the proposed method can perform well in the image fusion task and also help improve the performance of target detection.

## 5. Conclusions

In this paper, we propose a new end-to-end network to solve the problem of infrared and visible image fusion. For the characteristics of two different modal images, we design dual-stream asymmetric branched paths to extract infrared and visible image features, use the transformer to capture global information and long-distance dependencies in infrared images, and use residual dense blocks to fully extract texture details in visible images. Finally, the captured features are fully merged by the main path to further retain important information. This approach enables the preservation of both texture details from visible images and thermal radiation targets from infrared images in a superior manner. We conducted a large number of comparison experiments and generalization experiments testing using the RoadScene datasets and TNO datasets. The experimental results reveal that our approach outperforms existing techniques in both subjective and objective evaluations, demonstrating its outstanding performance and generalization ability. Target detection experiments were carried out on the MFNet datasets to showcase the prowess of our approach in elevating high-level visual tasks. In our future research, we will further modify our network details and design more appropriate loss functions to improve our method. We will design our method to be a unified fusion model and apply it to other multimodal image fusion fields, such as medical image fusion, remote sensing image fusion, and other tasks.

## Figures and Tables

**Figure 1 sensors-23-07097-f001:**
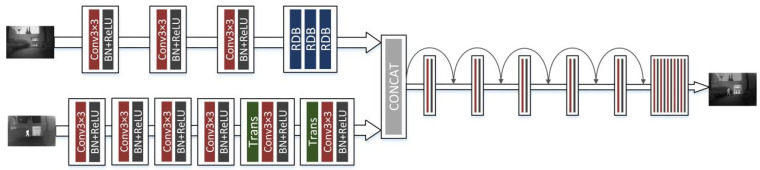
DSA-Net framework structure.

**Figure 2 sensors-23-07097-f002:**
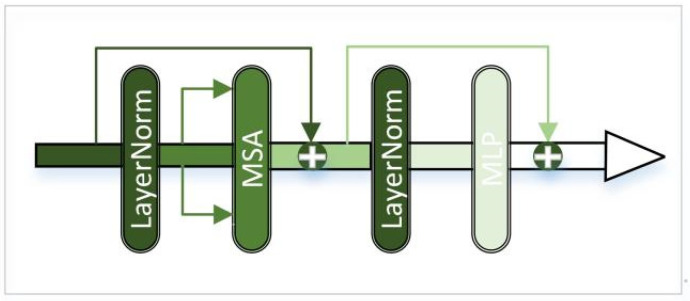
Transformer framework structure.

**Figure 3 sensors-23-07097-f003:**
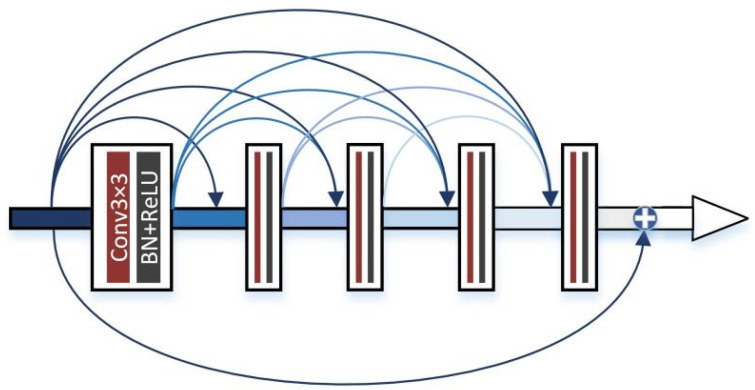
RDB structure details.

**Figure 4 sensors-23-07097-f004:**
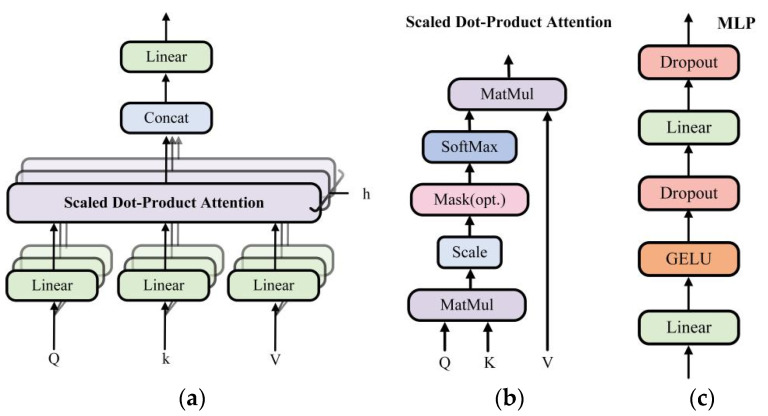
Transformer’s component structure. (**a**) MSA structure details; (**b**) Scaled Dot–Product Attention structure details; (**c**) MLP structure details.

**Figure 5 sensors-23-07097-f005:**
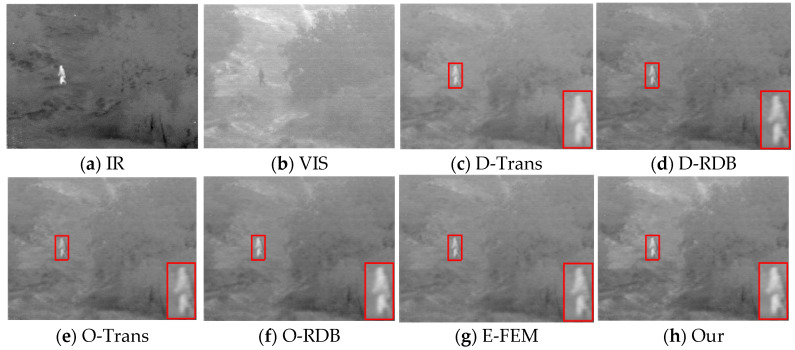
Subjective results of the ablation experiment TNO datasets.

**Figure 6 sensors-23-07097-f006:**
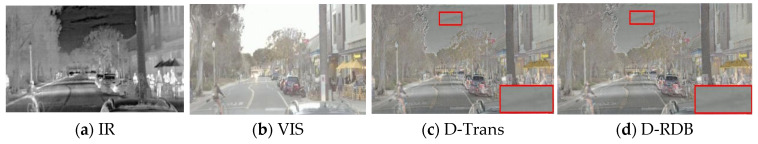
Subjective results of the ablation experiment RoadScene datasets.

**Figure 7 sensors-23-07097-f007:**
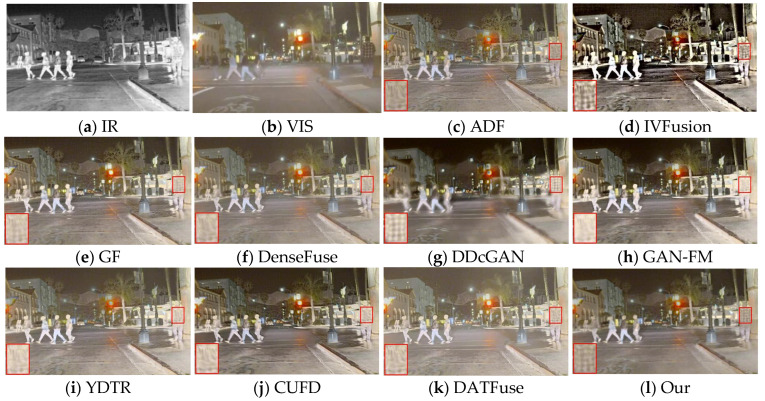
Fusion results of different methods for “nighttime on the road”.

**Figure 8 sensors-23-07097-f008:**
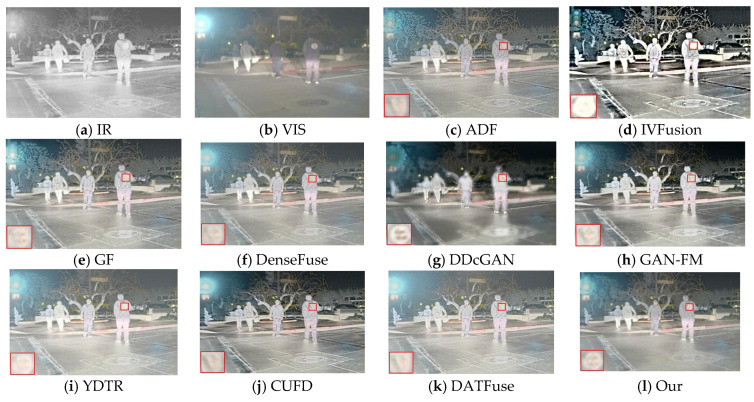
Fusion results of different methods for “nighttime sidewalks”.

**Figure 9 sensors-23-07097-f009:**
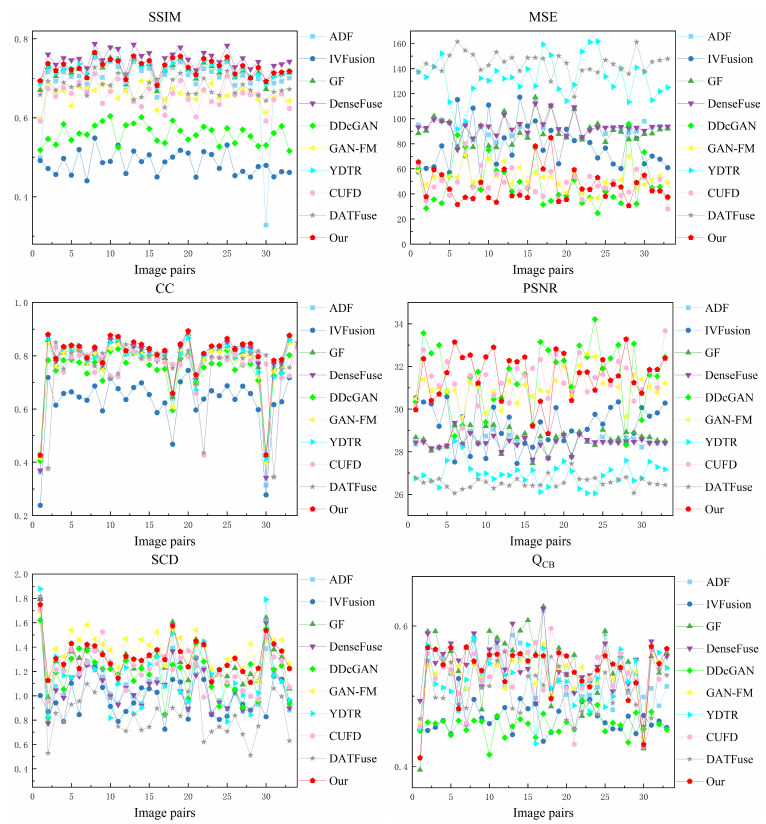
Performance graph of different fusion methods on RoadScene dataset.

**Figure 10 sensors-23-07097-f010:**
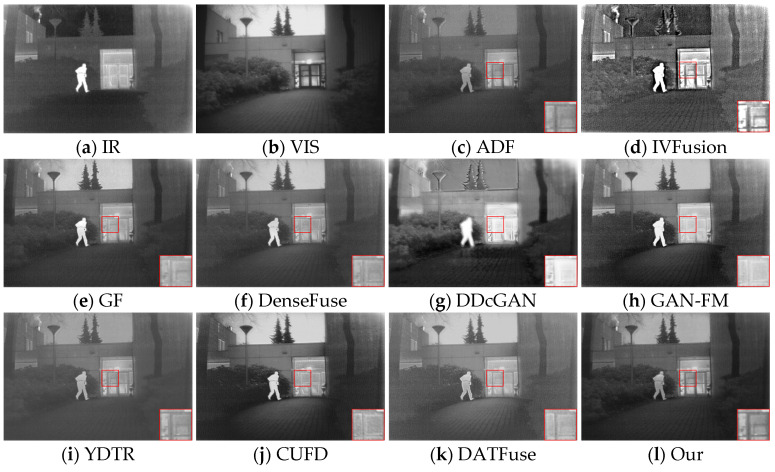
Fusion images of “person walking” for different methods, along with their respective source images.

**Figure 11 sensors-23-07097-f011:**
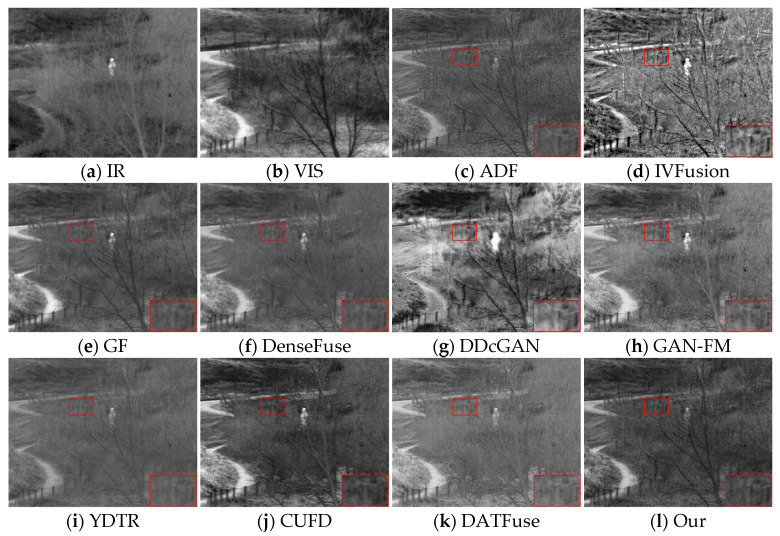
Fusion images of “forest trail” for different methods, along with their respective source images.

**Figure 12 sensors-23-07097-f012:**
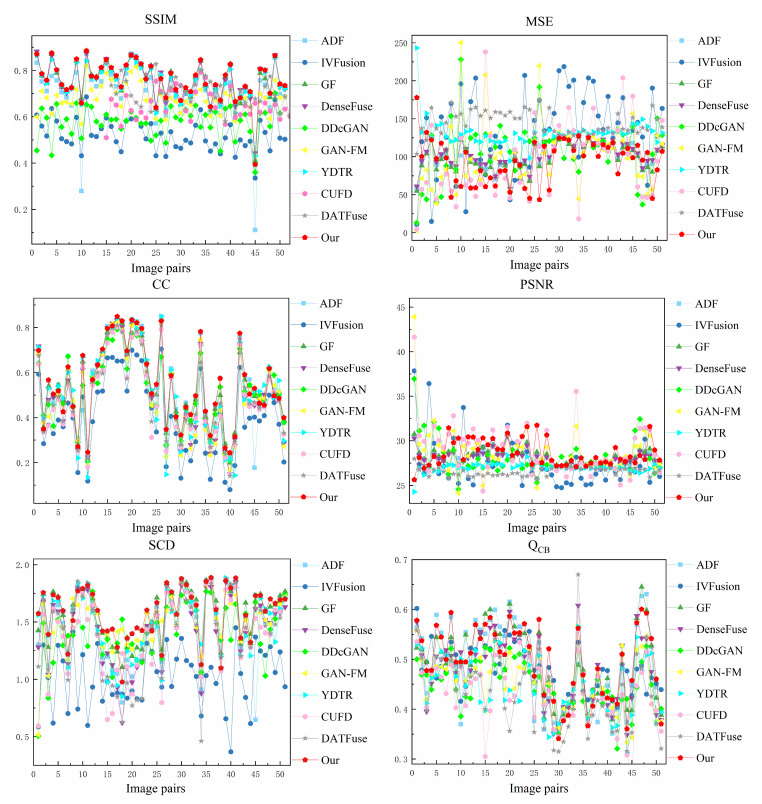
Performance graph of different fusion methods on TNO dataset.

**Figure 13 sensors-23-07097-f013:**
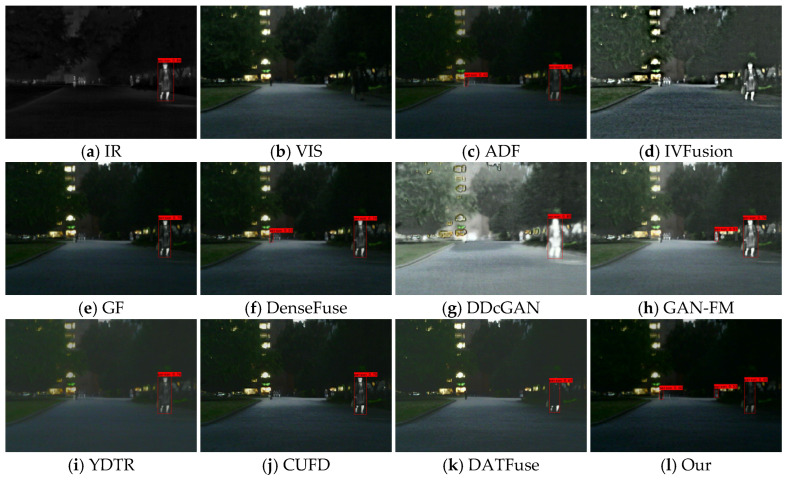
Subjective results of target detection on scene 00004N.

**Figure 14 sensors-23-07097-f014:**
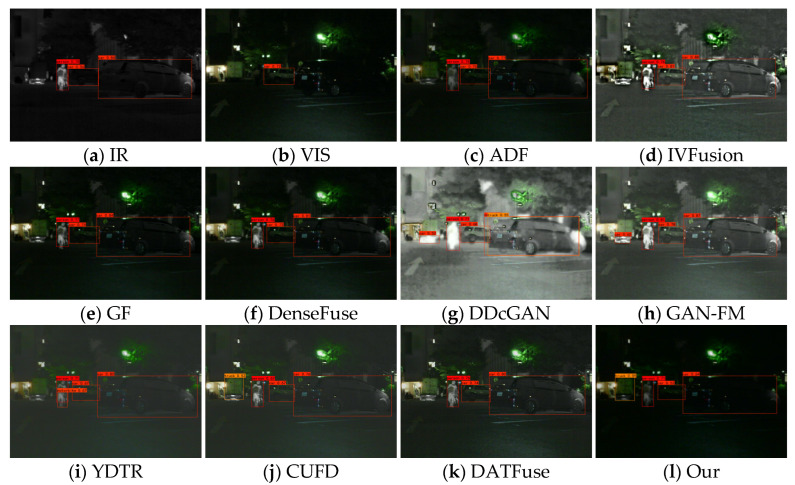
Subjective results of target detection on scene 00726N.

**Figure 15 sensors-23-07097-f015:**
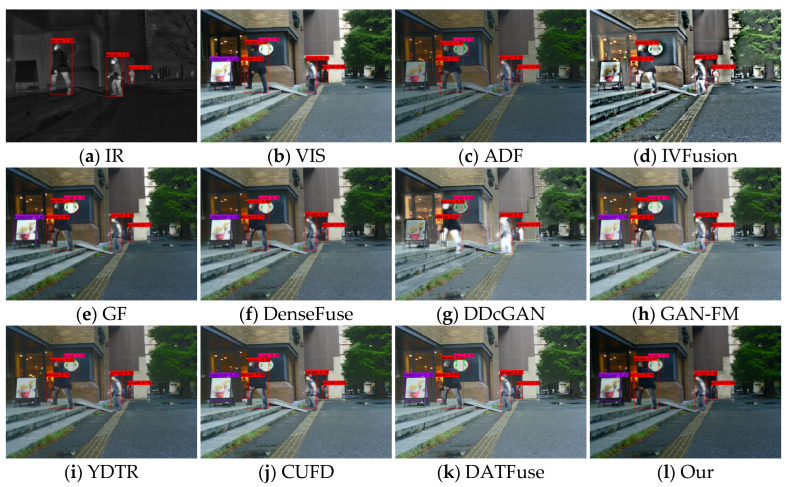
Subjective results of target detection on scene 00420D.

**Table 1 sensors-23-07097-t001:** Objective results of the ablation experiment RoadScene and TNO datasets.

Datasets	Methods	Quality Metrics					
		SSIM	MSE	CC	PSNR	SCD	Q_CB_
RoadScene	D-Trans	0.7141	125.1179	0.7754	27.1705	1.2882	0.4773
	D-RDB	0.7317	72.9455	0.7762	29.5688	1.2598	0.5086
	O-Trans	0.7306	68.2725	0.7837	29.8866	1.2907	0.5027
	O-RDB	0.7192	66.7998	0.7732	30.0231	1.3115	0.5112
	E-FEM	**0.7330**	82.1417	0.7815	29.0312	1.2557	0.5021
	Our	0.7277	**46.6912**	**0.7990**	**31.5830**	**1.3218**	**0.5469**
TNO	D-Trans	0.7125	117.3409	0.5523	27.4721	**1.5780**	0.4768
	D-RDB	0.7289	83.4061	0.5416	29.1615	1.4354	0.4805
	O-Trans	0.7539	83.2827	0.5279	29.1897	1.4338	0.4892
	O-RDB	0.7605	83.8169	**0.5553**	29.1480	1.5119	0.4945
	E-FEM	**0.7624**	90.1372	0.5351	28.7422	1.4452	0.4844
	Our	0.7587	**80.6446**	0.5452	**29.3554**	1.5180	**0.5000**

**Table 2 sensors-23-07097-t002:** Objective results of the comparative experiment RoadScene datasets.

Methods	Quality Metrics					
	SSIM	MSE	CC	PSNR	SCD	Q_CB_
ADF	0.6909	93.2764	0.7810	28.4483	1.0776	0.5285
IVFusion	0.4642	60.1766	0.6855	30.3363	0.9305	0.4535
GF	0.7190	90.6036	0.7769	28.5830	1.2868	0.5465
DenseFuse	**0.7453**	93.8907	0.7851	28.4170	1.0819	0.5440
DDcGAN	0.5589	48.4804	0.7410	31.5353	1.1833	0.4594
GAN-FM	0.6590	52.2970	0.7680	30.9994	**1.3848**	0.5327
YDTR	0.7231	131.9041	0.7771	26.9622	1.1619	0.5236
CUFD	0.6567	48.2179	0.7404	31.3925	1.2247	0.5307
DATFuse	0.6773	144.0092	0.7552	26.5545	0.8985	0.5072
Our	0.7277	**46.6912**	**0.7990**	**31.5830**	1.3218	**0.5469**

**Table 3 sensors-23-07097-t003:** Objective results of the generalization experiment TNO datasets.

Methods	Quality Metrics					
	SSIM	MSE	CC	PSNR	SCD	Q_CB_
ADF	0.7085	101.6540	0.5407	28.1317	1.4522	0.4888
IVFusion	0.5224	133.1557	0.4016	27.4738	1.0104	0.4795
GF	0.7529	99.9707	0.5368	28.2123	**1.5735**	0.4891
DenseFuse	0.7547	104.4317	0.5068	28.0037	1.4427	0.4774
DDcGAN	0.5785	107.4098	0.5120	28.2019	1.4089	0.4497
GAN-FM	0.6763	100.7944	0.5049	28.3592	1.5058	0.4540
YDTR	0.7443	131.2808	0.5213	26.9846	1.4979	0.4478
CUFD	0.6465	95.1249	0.4849	29.0752	1.3908	0.4291
DATFuse	0.7008	144.4359	0.5007	26.5583	1.4006	0.4343
Our	**0.7587**	**80.6446**	**0.5452**	**29.3554**	1.5180	**0.5000**

**Table 4 sensors-23-07097-t004:** Detection results of source images and different fusion methods.

Methods	mAP@0.5			mAP@0.9		
	Person	Car	Average	Person	Car	Average
IR	0.6307	0.3023	0.4665	**0.2562**	0.3013	0.2788
VIS	0.4953	0.7240	0.6096	0.1901	0.4358	0.3129
ADF	0.6935	0.7208	0.7072	0.2456	0.4505	0.3480
IVFusion	0.7288	0.7040	0.7164	0.1768	0.3733	0.2750
GF	0.6562	0.7300	0.6931	0.2415	0.4603	0.3509
DenseFuse	0.6915	0.7353	0.7134	0.2413	0.4425	0.3419
DDcGAN	0.4010	0.6968	0.5489	0.1072	0.3550	0.2316
GAN-FM	**0.7450**	**0.7548**	**0.7499**	0.2409	0.4178	0.3293
YDTR	0.7149	0.5708	0.6428	0.2348	**0.4972**	0.3660
CUFD	0.6637	0.6557	0.6597	0.2116	0.3730	0.2923
DATFuse	0.6794	0.7205	0.6999	0.2382	0.3958	0.3170
Our	0.7241	0.7388	0.7223	0.2458	0.4865	**0.3661**

## Data Availability

Source of dataset in experimental analysis: https://fifigshare.com/articles/TNOImageFusionDataset/1008029 (accessed on 17 March 2023), https://github.com/hanna-xu/RoadScene (accessed on 17 March 2023), and https://github.com/haqishen/MFNet-pytorch (accessed on 19 March 2023).

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
