# Peer review of "DSA-Net: Infrared and Visible Image Fusion via Dual-Stream Asymmetric Network"

_sensors, 2023, doi:10.3390/s23167097_

Round 1

Reviewer 1 Report

This paper proposes a dual-stream asymmetric fusion network with two different feature extraction networks for Infrared and Visible Images. The authors reported that the DSA-Net outperforms state-of-the-art methods in both qualitative and quantitative evaluations. The following problems need to be addressed:

1、Densely connected architectures, CNNs and transformer architectures are well known in the literature. Therefore, these portions cannot be considered as contributions. When the multi-head self-attention modules were examined, the transformer modules in previous studies were used in their original form. This part does not contain a contribution.

2、There are other studies in which all these architectures are used together, please do not think only in the field of Infrared and Visible Images. For this reason, I couldn't see any important novelty.

3、The author cites a sufficient number of references, but it is recommended to add more recent references in the past two years.

4、Figure 3, Figure 4, and Figure 5 suggest changing the arrangement and layout, and the parallel arrangement seems to be more beautiful and reasonable.

5、Among the nine comparative methods, there is only one method from 2022. It is recommended to supplement the experimental methods from the past two years to further verify the superiority of this method.

6、There are some Syntax errors, for example, in the sentence "In addition, we also apply the fusion results to the target detection task, which indirectly demonstrates the fusion performances of our method.", performance is Mass noun.

7、It is recommended to use underscores or other operations to highlight the second largest value in the table that represents objective comparison in the paper. For example, in Table 2, only one indicator can be highlighted as the optimal value, while all other indicators are the second largest value, but it cannot be clearly seen from the table.

Author Response

Dear Reviewers,

We highly appreciate the detailed valuable comments on our manuscript of “DSA-Net: Infrared and Visible Image Fusion via Dual-stream Asymmetric Network”. The suggestions are quite helpful for us and we have incorporated them in the revised paper. During the last few days, we have made careful major revisions on this paper.

As below, on behalf of my co-authors, I would like to elucidate all the points raised. And we hope you will be satisfied with our responses to the comments and the revisions for the previous manuscript.

Thanks and Best Regards!

Yours Sincerely,

Bin Yang

Reviewer 2 Report

The current paper proposes a methodology and, correspondingly, a Dual-stream Asymmetric Network architecture (DSA), in order to fuse visible and infrared images in optimum manner.  The DSA network is based on both CNNs and transformers. The paper is well written, demonstrating an increased technical and scientific quality, however, the following suggestions should be still considered:

(1.) The state of the art should be presented in a more extended manner,  while the corresponding limitations and the original contributions of the authors with respect to the state-of-the-art should be more clearly emphasized.

(2.) The TNO initials corresponding to the TNO dataset should be explained.

(3.) Figure 9 and Figure 10 have the same title (caption), so either they should be unified under the same title, or they should have different titles.

(4.)  The authors should also propose specific modalities in order to further improve the currently obtained results.  

In the current article, the authors demonstrated a very good English language level, however, minor editing issues still can be detected.

Author Response

(The authors gave the same response as above.)

Reviewer 3 Report

This paper presented a novel end-to-end network for infrared and visible image fusion, which combines CNN and Transformer for better feature extraction and image fusion. The work is interesting. The presentation is good and can be easily understood. The experiments are sufficient to demonstrate the effectiveness of the design. My minor comments are:

 1.     In the introduction, the literature review on deep learning for image fusion has to be improved as only two works were mentioned. After improving the literature review, the authors need to formulate better what the existing problems are and how the authors solve these problems.

2.     There are a few typos in the paper, the authors need to proofread the paper. For instance, line 61: “differ modal features”.

There are a few typos in the paper, the authors need to proofread the paper.

Author Response

(The authors gave the same response as above.)

Round 2

Reviewer 1 Report

The authors have answered my questions.